# Proof-of-Concept Organ-on-Chip Study: Topical Cinnamaldehyde Exposure of Reconstructed Human Skin with Integrated Neopapillae Cultured under Dynamic Flow

**DOI:** 10.3390/pharmaceutics14081529

**Published:** 2022-07-22

**Authors:** Irit Vahav, Maria Thon, Lenie J. van den Broek, Sander W. Spiekstra, Beren Atac, Gerd Lindner, Katharina Schimek, Uwe Marx, Susan Gibbs

**Affiliations:** 1Department of Molecular Cell Biology and Immunology, Amsterdam UMC Location Vrije Universiteit Amsterdam, Boelelaan 1117, 1081 HV Amsterdam, The Netherlands; 2TissUse GmbH, Oudenarder Str. 16, 13347 Berlin, Germany; 3Amsterdam Movement Sciences, Tissue Function & Regeneration, 1081 HV Amsterdam, The Netherlands; 4Amsterdam Institute for Infection and Immunity, Inflammatory Diseases, 1081 HV Amsterdam, The Netherlands; 5Department of Biotechnology, Technische Universität Berlin, Gustav-Meyer-Allee 25, 13355 Berlin, Germany; 6Provio GmbH, Oranienburger Chaussee 2, 16548 Glienicke/Nordbahn, Germany; 7Department of Oral Cell Biology, Academic Centre for Dentistry Amsterdam (ACTA), University of Amsterdam and Vrije Universiteit Amsterdam, 1081 LA Amsterdam, The Netherlands

**Keywords:** reconstructed human skin, hair follicle, in vitro, neopapillae, organ on chip, robust, reproducible, sensitizer

## Abstract

Pharmaceutical and personal care industries require human representative models for testing to ensure the safety of their products. A major route of penetration into our body after substance exposure is via the skin. Our aim was to generate robust culture conditions for a next generation human skin-on-chip model containing neopapillae and to establish proof-of-concept testing with the sensitizer, cinnamaldehyde. Reconstructed human skin consisting of a stratified and differentiated epidermis on a fibroblast populated hydrogel containing neopapillae spheroids (RhS-NP), were cultured air-exposed and under dynamic flow for 10 days. The robustness of three independent experiments, each with up to 21 intra-experiment replicates, was investigated. The epidermis was seen to invaginate into the hydrogel towards the neopapille spheroids. Daily measurements of lactate dehydrogenase (LDH) and glucose levels within the culture medium demonstrated high viability and stable metabolic activity throughout the culture period in all three independent experiments and in the replicates within an experiment. Topical cinnamaldehyde exposure to RhS-NP resulted in dose-dependent cytotoxicity (increased LDH release) and elevated cytokine secretion of contact sensitizer specific IL-18, pro-inflammatory IL-1β, inflammatory IL-23 and IFN-γ, as well as anti-inflammatory IL-10 and IL-12p70. This study demonstrates the robustness and feasibility of complex next generation skin models for investigating skin immunotoxicity.

## 1. Introduction

Pharmaceutical and personal care industries require human-like models for testing to ensure the safety of their products. Over the last years, this has led to the development of more and more complex single organ models and now it is necessary to combine these organoid models to investigate systemic effects and multiple organ interactions. This requires incorporating these next generation organoids into bioreactors containing microfluidics which will mimic vasculature flow and make it possible to link different organoids under defined conditions [1]. Our body barriers (skin, gastro-intestinal tract, lungs) provide the first route of exposure to substances from the environment, be it a drug, consumer product or toxic substance, and therefore such platforms should always incorporate one of these organs.

Skin tissue engineering is leading in this field, with the European Union (EU) forbidding cosmetics testing in living animals (2013), creating a huge driving force [2]. The ban called for the rapid development of human skin equivalents for efficacy and cytotoxicity testing, resulting in the reconstruction of human skin models that more and more closely represent human skin physiology [3]. As the skin provides a barrier between the human body and its external environment, it is continuously exposed to environmental chemicals, cosmetics and drugs which penetrate the skin through the stratum corneum and the hair follicles.

Relatively simple to complex skin models have been described which are grown under static culture conditions with the culture medium reaching the skin from underneath and the surface being exposed to air to promote optimal epidermal differentiation and the formation of a stratum corneum [4,5,6,7,8]. These models are starting to include the epidermis containing, for example, keratinocytes, melanocytes, Langerhans Cells, the dermis, which contains fibroblasts and endothelial cells, and, in a few cases, adipocytes [9,10,11,12,13]. Although there are various commercially available skin models that are currently being utilized for hazard assessment [11], there are no reconstructed human skin models (RhS) that include the hair shaft penetration route in addition to the stratum corneum. Furthermore, these complex skin models have yet to be incorporated into bioreactors which will enable perfusion and the dynamic flow of culture medium representative of blood capillary flow, with the perspective of linking these skin models to other organoids.

A number of reports describe skin models being incorporated into microfluidics bioreactors. A positive effect of microfluidic perfusion on RhS epidermal differentiation and morphology was observed when compared to static cultures e.g., the higher expression of epidermal differentiation markers and a lower number of proliferating epidermal cells, similar to the native human skin [14]; a reduction in tight junction integrity in epidermal keratinocytes cultured under dynamic flow was observed as a response to UV irradiation [15]; skin toxicity studies describe decreased barrier integrity after topical exposure to a benchmark irritant (0.2% sodium dodecyl sulphate) [16]; and the systemic toxicity effect of an anticancer drug (doxorubicin), which when applied to the circulating medium recapitulated the clinical cytotoxic effect as shown by the inhibition of basal keratinocyte proliferation in the skin constructs [17]. These above-mentioned studies demonstrate the potential of microfluidic systems to facilitate in vitro skin barrier models by reproducing a more physiological culture environment. However, until now, only relatively simple RhS models have been incorporated into the different bioreactors. These RhS have the epidermal stratum corneum as the barrier to the external environment, but lack other major skin penetration routes such as the hair shaft. The lack of such a major penetration route prevents the recapitulation of the physiological events involved in substance absorption in the skin [18,19,20,21], which becomes especially problematic when the epidermal barrier function is evaluated in topical application studies.

Recently we described epidermal invagination towards neopapillae (NP) (dermal papilla equivalents), incorporated into RhS. These NP formed cellular structures that are present in the initial stages of hair follicle morphogenesis and hair shaft formation. The initiation of follicle growth and consequently hair shaft formation is an important step towards the generation of an additional route for penetration of actives applied topically to the skin [18,19,20,21]. This RhS-NP model was cultured under static conditions (no dynamic flow) for 10 days, during which time epidermal invagination towards NP was observed. Invaginating keratinocytes exhibited a differentiation profile similar to the epithelial layers of the hair follicle, with keratin 15 and keratin 10 being expressed in the outer and inner keratinocyte layers, respectively. The inductive phenotype of the NP was observed by elevated gene expression of the signaling molecules Wnt5a, Wnt10b and LEF1, which are characteristically elevated in the early stages of hair follicle morphogenesis. The next step would be to incorporate dynamic flow, mimicking capillary pressures, into this RhS-NP model in order to enable it to be linked with organoids representing our internal organs in the future in order to investigate the systemic toxicity effects of substances breaching the skin barrier. The aim of this study was to obtain robust culture conditions for culturing RhS-NP in a microfluidics bioreactor, HUMIMIC Chip2 (TissUse, Berlin, Germany), for 10 days. Attention was paid to obtaining extensive intra-experimental and inter-experimental information on the viability (lactate dehydrogenase release which is indicative of porous cell membranes and cytotoxicity) and metabolic state (glucose uptake from the culture medium) of the cultures. Since it is most important that the RhS culture under dynamic flow responds to test substances at least as well as the current static models, the sensitizer cinnamaldehyde was topically applied to RhS and the release of multiple (pro- and anti-)inflammatory cytokines, including the sensitizer biomarker IL-18 [22], and lactate dehydrogenase (LDH) was determined in the circulating culture medium.

## 2. Materials and Methods

### 2.1. Human Tissue and Cell Culture

The cell culture was as described previously [23]. Keratinocytes and dermal fibroblasts were isolated from human juvenile foreskin which was obtained as rest material after routine circumcisions. Dermal papilla cells were isolated from occipital and temporal follicular units in human scalp skin, and contained mostly growing anagen VI hair follicles. Follicular units were obtained from disposed excess skin samples derived from adult male and female patients undergoing hair transplantation surgery. All skin samples were obtained with informed consent and ethical approval from the Ethics Committee of the Charité Universitätsmedizin, Berlin, Germany, in compliance with the relevant laws.

### 2.2. Neopapillae Construction

Neopapillae construction was as described previously [23]. In short, 3 × 10^4^ dermal papillae cells/cm^2^ were seeded into an ultra-low-attachment 6-well plate (Corning, NY, USA) containing neopapillae-medium (Dulbecco’s Modified Eagle’s Medium (DMEM) supplemented with 10% fetal bovine serum (Hyclone Laboratories Inc., Logan, UT, USA) and 1% penicillin/streptomycin (Corning, NY, USA) essentially as described previously [24]. Dermal papillae cells self-assembled into spheroids during six days of culture and were then further referred to as neopapapillae (NP). The dense spheroidal structure of the spheroids was observed throughout the bulk cultures. The neopapillae medium was refreshed on the third day of culture and NP were incorporated into RhS at day six.

### 2.3. Reconstructed Human Skin with Neopapillae (RhS-NP)

RhS-NP construction and all culture media are exactly as described previously [23]. RhS were generated in a Millicell^®^ insert of 9 mm inner diameter and 0.4 μm pore size (Merck, Darmstadt, Germany). The fibroblast (7 × 10^4^ cells/mL)—NP (approximately 90 neopapillae per gel) populated hydrogel (rat tail collagen) was pipetted into the Millicell^®^ insert. After one day of submerged culture in keratinocyte medium-I (DMEM/Ham’s F12 (Corning, NY, USA) in a 3:1 ratio supplemented with 5% fetal clone III (Hyclone, UT, USA), 1 μmol/L hydrocortisone (Sigma-Aldrich, St. Louis, MO, USA), 1 μmol/L isoproterenol hydrochloride (Sigma-Aldrich), 0.1 μmol/L insulin (Sigma-Aldrich), 2 ng/mL human keratinocyte growth factor (Sigma-Aldrich), and 1% penicillin/streptomycin), keratinocytes (6.5 × 10^4^ cells/hydrogel) were seeded on top of the hydrogel and cultured submerged for three to four days. Subsequently, RhS-NP were further cultured at the air-liquid interface for another one to three days in KC-II medium (DMEM/Ham’s F12 (Corning, NY, USA) in a 3:1 ratio containing 1% fetal clone III, 1 μmol/L hydrocortisone, 1 μmol/L isoproterenol hydrochloride, 0.1 μmol/L insulin, 10 μmol/L L-carnitine (Sigma-Aldrich, St. Louis, MO, USA), 0.01 mol/L L-serine (Sigma-Aldrich), 50 μg/mL ascorbic acid (Sigma-Aldrich), and 1% penicillin/streptomycin (Corning, NY, USA)) before incorporating into the HUMMIK Chip2 (TissUse, Berlin, Germany).

### 2.4. Incorporation of RhS-NP into the Multi-Organ-Chip

RhS-NP in standing culture inserts were placed in the big compartment of the HUMMIK Chip2, while the small compartment (reservoir) was used for medium change (Figure 1). RhS-NP were introduced to the HUMMIK Chip2 on days one to three of the air exposure culture as previously described [25,26,27]. After flushing the KC-II medium into the channels of the chip, 200 µL of the same medium was pipetted into the big compartment and 700 µL was added to the reservoir. RhS-NP were then carefully placed standing into the big compartment of the chip using forceps. This resulted in the culture insert standing directly onto the chip glass slide, while the medium level only reached the transwell membrane surface, thus ensuring the culture at the air liquid interface within the HUMMIK Chip2. The culture compartments were then hermetically sealed with screwball lids. The chips were then connected to the micro pumps (2.5 Hz/microfluidic circuit) and placed in the incubator at 37 °C, 5% CO_2_. 450 µL of the reservoir medium was refreshed daily. The RhS-NP were cultured air-exposed in the chip for 10–14 days under a dynamic flow to promote epidermal stratification and differentiation.

### 2.5. Topical Application of Cinnamaldehyde onto RhS-NP Cultured under Dynamic Flow

After 10 days of dynamic air-exposed cultivation in the chip, RhS-NP were removed from the chip and placed shortly in a 6-well plate with KC-II medium (1 mL/well). Meanwhile, the chips were flushed with KC-II medium without hydrocortisone to remove old medium residues (hydrocortisone is an immune suppressive medium component). The chip was then refreshed with KC-II, 700 µL was added to the reservoir and 200 µL was added to the culture compartment. Finn Chambers^®^ filter paper discs (8 mm) were impregnated with 25 µL cinnamaldehyde (10 µM, 20 µM, 40 µM, 60 µM, 80 µM). The vehicle was 1% DMSO in H_2_O. Excess fluid was removed and filters were placed topically onto the stratum corneum of RhS-NP. The vehicle controls contained 1% DMSO in H_2_O; RhS-NP controls were unexposed cultures without filter paper discs. RhS-NP were transferred back into the chip and further incubated for 24 h at 37 °C, with 5% CO_2_. After 24 h of cinnamaldehyde exposure, RhS-NP were harvested.

### 2.6. Histology

RhS-NP were fixed in 4% formaldehyde and embedded in paraffin. Paraffin sections (5 μm) were stained with hematoxylin and eosin (H&E) and mounted with Roti^®^-Histokitt solution (Roth). Tissue sections were photographed using a Nikon Eclipse 80i microscope (Düsseldorf, Germany) with NIS elements AR 3.2 software (Nikon, Tokyo, Japan).

### 2.7. Glucose and LDH Analysis of Culture Supernatant

The culture medium (450 µL) was collected daily and transferred to 96-well analysis plates. The supernatant for glucose, cytokine bead array and ELISA (enzyme-linked immunosorbent assay) assays was stored at −20 °C until further use, while the supernatant for LDH-measurement was stored up to four days at 4 °C. All samples were thawed overnight at 4 °C before analysis. 384-well plates, which were used for all measurements, were first treated with air plasma in order to create a hydrophobic plate surface. All samples and standards were measured in duplicates. Glucose and LDH amounts were determined in the sample as described by the kit suppliers using the Stanbio Glucose LiquiColor^®^ reagent kit (Stanbio Laboratory, Boerne, TX, USA) and Cytotoxicity Detection KitPLUS (Roche Diagnostics GmbH, Mannheim, Germany). All absorbance measurements were performed in a microplate-reader (Fluostar Omega, BMG Labtech, Ortenberg, Germany).

Glucose: samples were diluted 1:5 with glucose-free medium. Standard samples were prepared with two-fold serial dilutions, which resulted in 8-concentration points in the range of 0.215–2.295 g/L. The glucose-free medium was used as blank. The reagent (57 µL) and sample (3 µL) or standard were pipetted into the wells of a 384-well plate. Plates were shaken at 1100 rpm for 15 min at RT in the dark followed by a 1 min centrifugation at 100× *g*. The absorbance was measured at 520 nm with a plate reader and data was analyzed with MARS (BMG Labtech GmbH, Ortenberg, Germany) data analysis software.

LDH: Standard solutions were prepared by two-fold serial dilutions in PBS/1% BSA, covering the range of 10 mU/mL to 0.78125 mU/mL, where PBS/1% BSA was used as blank. Reagent (12.5 µL) and sample (12.5 µL) or standard were pipetted into the wells of a 384-well plate. The plates were shaken at 1100 rpm for 20 min at RT in the dark, then placed in a plate reader and the absorbance was read at 490 nm and 680 nm. Data was analyzed using the MARS data analysis software. LDH values were calculated as a percentage of the positive control, which was the total LDH release in a lysed RhS-NP sample.

### 2.8. Cytokine Analysis

IL-6, MCP1, IL-8/CXCL8 and IL-18: Enzyme-linked immunosorbent assays (ELISAs) were performed on RhS-NP culture supernatants. The ELISAs reagents were used in accordance with the manufacturer’s specifications. The required antibodies and recombinant proteins were supplied by R&D Systems, Inc. (Minneapolis, MN, USA) for IL-6 and MCP-1, except for IL-8, which was supplied by Diaclone SAS (Besançon, France) and IL-18, which was supplied by MBL (MBL International, Woburn, MA, USA). Absorbance was measured at 450 nm or 490 nm, using a microplate reader (ELISA reader Mithras LB 940, Berthold Technologies, Bad Wildbad, Germany). Data was analyzed using the MicroWin data analysis software.

IL-1β, IFN-α2, IFN-γ, TNF-α, MCP-1, IL-6, IL-8, IL-10, IL-12p70, IL-17A, IL-18, IL-23, IL-33: Supernatants were analysed with LEGENDplex™ Human Inflammation Panel 1 according to the manufacturer’s instructions (BioLegend, San Diego, CA, USA). Acquisition was performed on a Attune NxT Flow Cytometer (Thermo Fisher Scientific, Eugene, OR, USA), (excitation Y1 (561 nm; 50 mW), emission 585/16 nm; ex. R1 (637 nM; 100 mW) em. 670/14 nm). Analysis was performed in LEGENDplex™ Data Analysis Software Suite (BioLegend).

### 2.9. Statistical Analysis

Three independent experiments were carried out, each representing cells from a different donor with an intra-experiment triplicate. Data are presented as mean, or standard error of the mean (SEM) as indicated. The differences between groups were tested for statistical significance using a Shapiro-Wilk test to assess normality, then one-way ANOVA; one-way ANOVA (Kruskal-Wallis test) followed by Dunn’s multiple comparisons test; one-way ANOVA (Friedman test) using GraphPad Prism 9 software (GraphPad Software Inc., La Jolla, San Diego, CA, USA) as indicated in the figure legends. Differences were considered to be significant when *p* < 0.05.

## 3. Results

### 3.1. Histology of Reconstructed Human Skin with Integrated Neopapillae in the HUMIMIC Chip2

An extensive characterization and immunohistochemical description of the RhS-NP model cultured under conventional static conditions has been published in Vahav et al. [23]. The starting point of this study was to further develop the RhS-NP to enable construction and a stable culture within the HUMIMIC Chip. The diameter of the self-organized neopapillae spheroids used to incorporate into RhS was comparable to that of dermal papillae found in scalp hair and as described previously for those incorporated into static RhS-NP (mean diameter 97 µm; SEM ± 16.6 µm) (Figure 2a) [23].

When grown under dynamic flow for 10 days, starting from the period of air exposed culture, RhS developed a differentiated epidermis that included the basal, granular and cornified keratinocyte layers with multiple NP spheroids within the fibroblast populated hydrogel. Moreover, an epidermal invagination towards the NP, similar to that observed in static cultures, was visible in the dynamic cultured RhS-NP, indicating that the RhS-NP could be successfully integrated into the HUMIMIC Chip2. (Figure 2b–d).

### 3.2. Low Inter- and Intra- Experimental Variation in Glucose Consumption and LDH Release during Dynamic Flow Culture

In order to determine how robust the RhS-NP was when cultured under dynamic flow conditions for 13 days (experiment 1) and 10 days (experiments 2 and 3), inter-experimental variation between three independent repeat experiments was determined. Figure 3 shows the averaged values of up to 21 intra-experiment replicates at each time point for the 3 independent experiments. Glucose levels were determined during the 10-day culture period (for experiment 1 this was day 4–13). All RhS-NP displayed similar levels of glucose levels remaining in culture supernatants during the 10 day culture period, with a 10 day average value of 11.1 mmol/L (SEM ± 0.38), 8.8 mmol/L (SEM ± 0.53), 9.4 mmol/L (SEM ± 0.48) for the first, second, and third independent experiments, respectively.

The LDH release, representative of leaky cell membranes and cell death, was assessed in RhS-NP cultures throughout the 10 days in all independent experiments. The first repeat experiment showed a slightly higher 10 day average of LDH release (9.8%, SEM ± 1.10), while the second and third repeat experiments demonstrated almost identical average values of 7.0% (SEM ± 0.56) and 6.9% (SEM ± 1.07), emphasizing the reproducibility of the RhS-NP culture’s viability. Notably, in all cases, LDH values were very low, indicating very little cell death during the dynamic flow 10-day period.

Next, the intra-experimental variation of the 21 replicates within the third repeat experiment was intensively investigated. Since the RhS-NP would hereafter be exposed to different concentrations of cinnamaldehyde in triplicate, the RhS-NP batch were already divided into groups of three replicates. Extremely little intra-experimental variation was observed between the replicates within a single experiment with regard to LDH (20 replicates) and glucose consumption (21 replicates) (Figure 4). All RhS-NP displayed similar levels of glucose consumption with an average of 7.6 mmol/L per day (SEM ± 0.44) at day 10 under dynamic flow. LDH levels remained low with an average value of 2.70% (SEM ± 0.20) at day 10. Similar results were also observed on days one to nine (data not shown). Additionally, in the day 10 culture supernatants, the release of cytokine IL-18 was determined in order to identify the base line secretion in RhS-NP, since this cytokine is expected to increase upon sensitizer exposure. In the past we have shown that intracellular stored IL-18 is produced only upon contact sensitizer exposure and not upon irritant or respiratory sensitizer exposure, and that this intracellular IL-18 is released into culture supernatants when the cell membrane becomes leaky due to cytotoxicity [22]. RhS-NP released stable, low amounts of IL-18 in the range of 6.7–26.4 pg/mL with an average of 14.5 pg/mL (SEM ± 1.2) (Figure 4c).

Taken together, these findings indicate that the static RhS-NP culture had successfully been transferred to dynamic flow in the HUMIMIC Chip2 and that robust reproducible base line data was achieved within an experiment and between experiments.

### 3.3. Topical Application of Sensitizer Cinnamaldehyde on RhS-NP

After 10 days of dynamic flow culture, RhS-NP from experiment 3 were topically exposed via the stratum corneum to increasing concentrations of cinnamaldehyde or its vehicle, and further cultured under dynamic flow for 24 h (Figure 5a). Histological analysis of RhS-NP after cinnamaldehyde exposure showed clear disruption of the upper epidermal layers including the invaginating epidermis reaching towards the neopapillae (cf. Figure 2b–d and Figure 5b).

Lactate dehydrogenase is only detected in culture supernatants when cell membranes become leaky due to cell toxicity. As observed in Figure 5c, LDH levels remain very low and comparable in both the unexposed (no filter paper) and vehicle (1% DMSO in water) impregnated filter paper disc, indicating that the addition of the filter paper disc and vehicle is not cytotoxic to RhS-NP. The LDH release increased in a dose dependent manner when the three highest concentrations were applied, reaching an average of 20% cytotoxicity with 80 mmol/L cinnamaldehyde. A similar dose dependent increase was observed in IL-18 release, with the significant increase in IL-18 confirming that the RhS was correctly scoring cinnamaldehyde as a sensitizer [22] (Figure 5c).

The release of (pro- and anti-)inflammatory cytokines including chemokines into the microfluidics compartment after topical exposure to cinnamaldehyde was further investigated. The slightly cytotoxic cinnamaldehyde concentration of 80 mM was not included in this extended analysis. Of the 12 additional cytokines studied, eight were detectable within the limits of the kit specifications (Table 1). Whereas inflammatory cytokines IL-6, IL-8 and MCP-1 showed no change in levels compared to vehicle exposed RhS-NP, pro-inflammatory cytokine IL-1β, inflammatory cytokines IL-23 and IFN-γ, and anti-inflammatory cytokines IL-10 and IL-12p70 showed a dose dependent trend to increased levels in culture supernatants. Due to two of the 21 RhS-NP being outliers for some of the cytokines for unknown reasons, statistical significance was not reached (Figure 6).

## 4. Discussion

Our results show that the complex static RhS-NP culture can be successfully transferred to the dynamic flow in the HUMIMIC Chip2 and that robust base line data can be achieved within an experiment and between experiments. We show that the full thickness RhS-NP can be cultured for at least 10 days under dynamic flow which mimics flow within our blood capillaries. Our proof-of-concept study showed toxicity testing with the skin sensitizer cinnamaldehyde resulting in pro-, anti- and inflammatory cytokine release into the microfluidics compartment.

Importantly, our study focusses on intra- and inter- experiment reproducibility in a complex skin on chip model. Three independent experiments were performed, each with up to 21 intra-experiment replicates. This set up was used anticipating experiments where a high number of replicates within a dose response study of a test substance is required. Our results show that within and between experiments very little variation is observed, indicating that the dynamic culture in the HUMIMIC Chip2 is a robust standardized method capable of producing reproducible results for at least 10 days, which was the duration of this study. When compared to other studies in the literature (as discussed below), intra- and inter experiment reproducibility is generally not discussed, and the number of replicates is often not mentioned or is considerably lower [14,15,28,29]. The emphasis of many studies is on imaging the organoids and vasculature, and obtaining results from testing a substance rather than in providing evidence on the robustness of the platform. Furthermore, our RhS-NP is more complex than other skin organoid models incorporated into the chips since it also contains the precursor of the hair shaft, which is a major substance penetration route in the skin.

A number of studies describe vascularizing chips since the vascularization of skin models is crucial for oxygen and nutrient supply to all cellular components within the construct [30,31,32]. Microvascular structures have been introduced into perfused skin models by using three dimensional (3D) printing techniques. These studies showed printed channels in skin models which were coated with endothelial cells and demonstrated endothelial barrier function in permeability assays using test molecules [33,34]. The incorporation of a vascularized network into the HUMIMIC microfluidic device, similar to the device used in this study, was achieved in a previous study by coating the microfluidic channels with human endothelial cells that responded by aligning in the direction of fluid flow within the chip [35]. A fibrin scaffold with adipose stromal- and endothelial cells was incorporated into the culture compartment and microvascular-like structures formed which remained stable for two weeks of dynamic culture [36]. These studies highlight the potential of introducing such a vascular network into RhS-NP in future studies using the HUMIMIC platform.

Our study is a stepping stone towards future multi-organ on chip platforms which interconnect complex barrier organoids and internal organoids required to adequately mimic human physiology and disease. Microfluidic platforms containing several culture compartments interconnected with micro channels already exist [37,38]. These platforms are showing potential as a prediction tool for compound metabolism in target organs and in pharmacokinetic studies. For example, the two organ chip platform (HUMIMIC Chip2) has been used for the dynamic culture of human liver equivalents with either human skin biopsies or intestine equivalents showcasing the native exposure routes to target organs after troglitazone exposure [39]. The same microfluidic platform was used for the co-culture of EpiDerm™ (reconstructed human epidermis) and liver spheroids, demonstrating the effect of different exposure regimes (single or repeated application of a topical or systemic exposure) on the pharmacokinetics and pharmacodynamics of the tested chemicals (hyperforin and permethrin) [40]. De Mello et al. demonstrated how a topical application to a synthetic skin surrogate in a heart-liver on-a-chip platform can predict the effect of transdermal drug exposure on liver and heart tissue functions [41]. Recently, we described a multi-organ-on-chip approach, showing the topical exposure of oral mucosa to metals which are known to activate the immune system and which in turn may result in an allergic rash and inflammation of the skin [42,43]. Reconstructed human gingiva and reconstructed human skin containing MUTZ-3 derived Langerhans cells integrated in the epidermis (RHS-LC) were incorporated into a HUMIMIC Chip3plus connected by dynamic flow. After an initial culture period of 24 h to enable stable dynamic culture conditions to be achieved, the sensitizer nickel sulphate was applied topically to the gingiva for 24 h and LC activation (maturation and migration) was determined in RHS-LC after an additional 24 h incubation time. Nickel exposure resulted in increased activation of LC as observed by increased mRNA levels of CD1a, CD207, HLA-DR and CD86 in the dermal compartment (hydrogel of RHS-LC (PCR)). This is the first study to describe systemic immunotoxicity and immune cell activation in a multi-organ setting.

In our current study, topical cinnamaldehyde exposure to RhS-NP cultured under dynamic flow resulted in dose-dependent cytotoxicity (increased LDH release) and elevated cytokine secretion of contact sensitizer specific IL-18, pro-inflammatory IL-1β, inflammatory IL-23 and IFN-γ, as well as anti-inflammatory IL-10 and IL-12p70. Only increased IL-18 release was significant with trends to increased cytokine release being observed for the other cytokines. This was most probably due to the number of replicates studied (three replicates) and because two outlier values were observed in the dose responses. Of note, these outliers were not consistent between all cytokines studied and were not related to the same RhS-NP sample; neither did these RhS-NP cultures show any variation in LDH release or glucose uptake in the time period before exposure (Figure 4), and therefore the reason for the deviance remains unknown. Since the focus of this study was to demonstrate intra- and inter- experiment variation, these outliers were clearly represented in this study. However, upon further assay standardization, criteria could be reached to eliminate such outliers if necessary (e.g., if two out of three runs are consistent, the outlier may be eliminated from the final result interpretation if clearly indicated). By using LEGENDplex™ Human Inflammation Panel 1 (BioLegend), 13 different cytokines could be analysed in only 10 µL of culture supernatant, thus overcoming the limitations of the small volumes typically obtained in microfluidic chips. However, the levels of three cytokines (MCP-1, IL-6 and IL-8) were out of range (too high), and therefore amounts had to be quantified by ELISA and the levels of four cytokines (IFN-α2, TNF-α, IL-17A, IL-33) were below the detection limit of the assay.

Limitations: In the current study, the RhS-NP were removed from the HUMMIC chip2 in order to ensure that the system could be thoroughly flushed with hydrocortisone free medium to avoid a possible hydrocortisone-induced immune suppression of cytokine release. This opportunity was taken to topically apply the cinnamaldehyde before replacing RhS-NP into the chip. Whether this refreshment with hydrocortisone free medium is necessary has yet to be determined. If it is not, then in the future the RhS-NP could remain in the chip during chemical administration, meaning additional handlings. The main limitation of our study and studies by others is the sample size. In our study, RhS-NP of 9 mm diameter were used to incorporate into the HUMMIC chip2 and total culture medium circulating the microfluidics and reservoir was 900 µL. Tissue histology can only be assessed when cultures are harvested and so advancements in, for example, 2-photon microscopy which enables live imaging, would greatly contribute to the field. Furthermore, the development of real time sensors and probes to automatically quantify LDH, glucose and inflammatory mediators (cytokines) is required to reduce the significant hours involved in sampling and analyzing, as well as minimizing the volume of the culture medium required to perform the assays. These advancements would make larger scale implementation more feasible in the future. Until these are fully implemented, the use of multiplex panels to assess cytokines and metabolites (protein and RNA) can be used.

Future perspectives: When comparing the results obtained from the current study with our previous results obtained from static RhS culture, a similar degree of robustness was observed. Both static and dynamic cultures exhibit a stratified and differentiated epidermis on a fibroblast populated hydrogel and secrete cytokines when exposed to chemicals. In both models, NP and functional LC can be included into the RhS [22,23,42,44]. However, both static and dynamic RhS models currently have the same limitation in that, as with all in vitro skin models, the barrier competency is slightly impaired due to the humid conditions within the culture incubator (95% humidity). It may be possible in the future to introduce an air flow into the upper air exposed compartment of the dynamic RhS in order to reduce humidity, dry the stratum corneum, stimulate desquamation and in turn improve barrier competency, which would be an important step forward for dermal penetration studies and bioavailability. The future of next generation risk assessment lies in developing more human physiologically representative organoid cultures which can be integrated into multi-organ platforms. The HUMIMIC Chip2 bioreactor easily facilities the multi-organ culture enabling such future systemic toxicity studies.

## Figures and Tables

**Figure 1 pharmaceutics-14-01529-f001:**
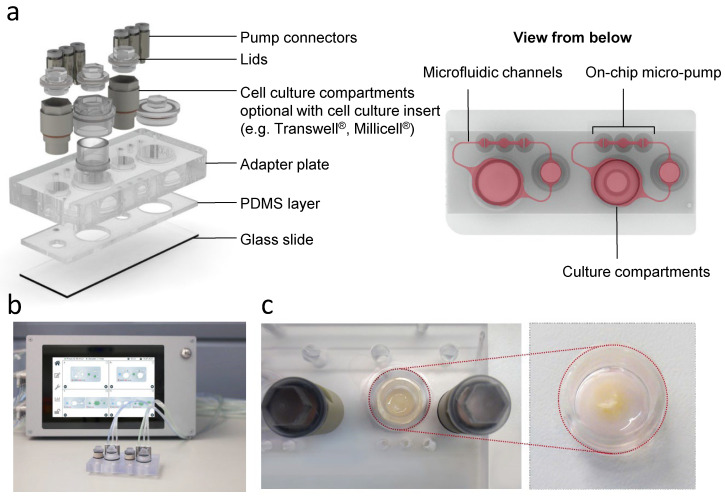
Reconstructed human skin with neopapillae (RhS-NP) integrated in the HUMIMIC Chip2. (**a**) The left image shows individual layers of the HUMIMIC Chip2 device: a glass slide with PDMS layer imprinted with microfluidic channels and an adapter plate containing the cell culture compartments and pump connectors of two independent circuits for replicate culture within a single chip. The right image exhibits the HUMIMIC Chip2 from below: the small compartment is the reservoir for the culture medium and the big compartment is for RhS-NP. The two compartments are interconnected with microfluidic channels filled with culture medium. (**b**) macroscopic view of the HUMIMIC Chip2 with on-chip micropumps, generating a pulsatile flow, adjusted by a control unit. (**c**) RhS-NP in Millicell^®^ cell culture inserts (9 mm inner diameter) integrated in the HUMIMIC Chip2 device (RhS-NP in culture insert marked in red).

**Figure 2 pharmaceutics-14-01529-f002:**
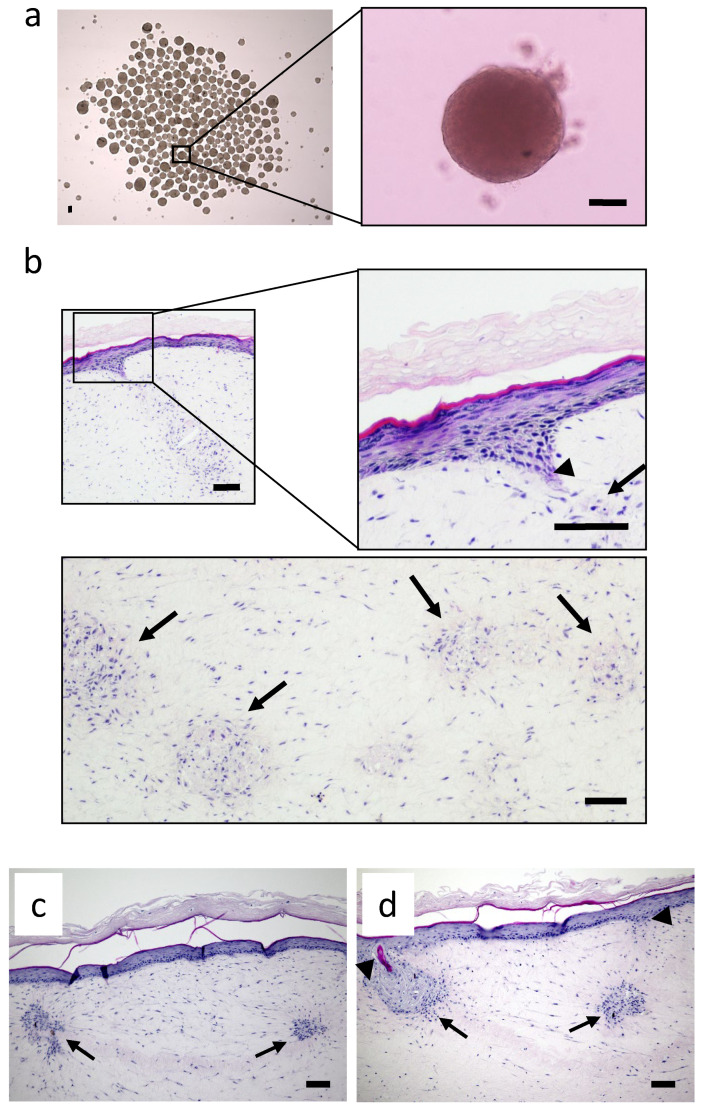
RhS-NP cultured under dynamic flow. (**a**) light microscopy image of bulk culture of neopapillae spheroids (inlay shows one spheroid) cultured in a 6-well plate, scale bar = 100 µm; (**b**–**d**) hematoxylin and eosin-(H&E) stained paraffin sections show NP located beneath the epidermis within the hydrogel (arrows) and invaginating epidermal keratinocytes in the vicinity of an NP (arrow heads). Scale bar = 100 µm.

**Figure 3 pharmaceutics-14-01529-f003:**
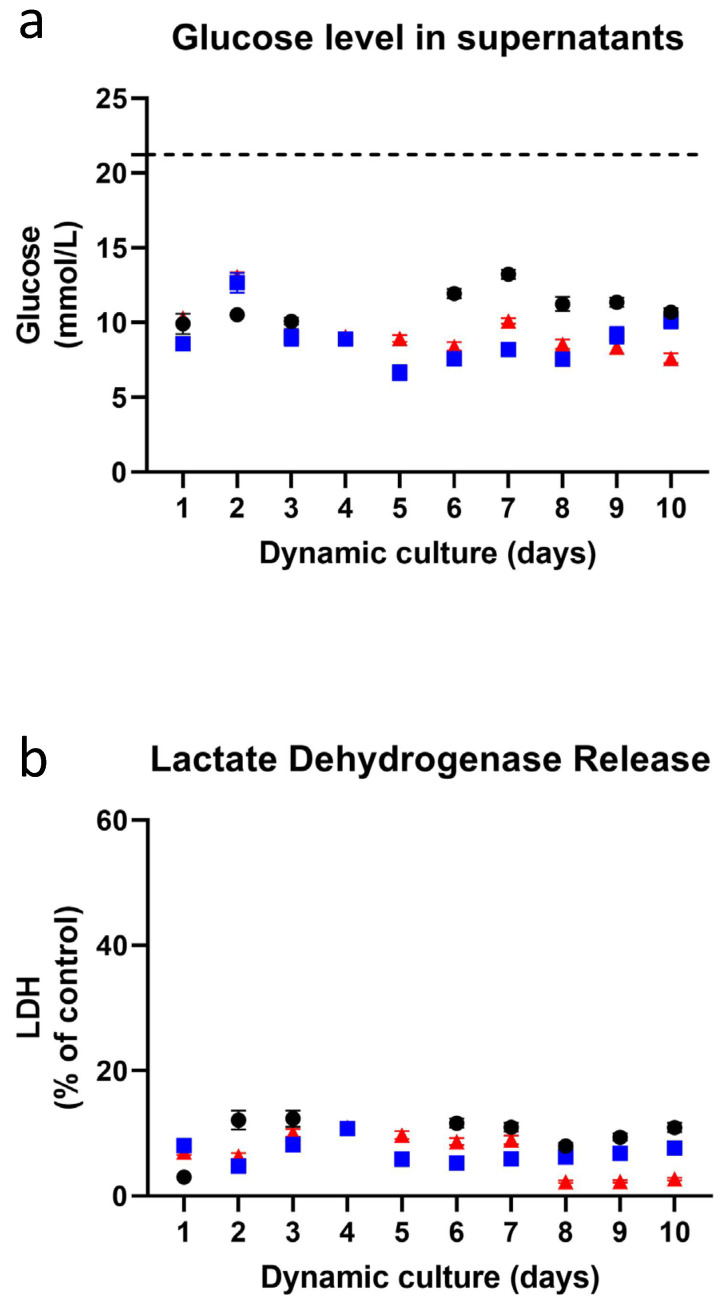
Inter-experiment reproducibility. (**a**) Glucose level in supernatants (glucose concentration in original medium is shown in a dashed line) and (**b**) lactate dehydrogenase (LDH)-release levels were measured daily from the supernatant of RhS-NP cultured over 10 days under dynamic flow. The data represents three independent experiments (experiment 1: black dots; experiment 2: blue squares; experiment 3: red triangles) and values are expressed in mean ± SEM of up to 21 intra-experimental replicates. A Shapiro-Wilk test was used to assess normality, and no significant difference was found with one-way ANOVA.

**Figure 4 pharmaceutics-14-01529-f004:**
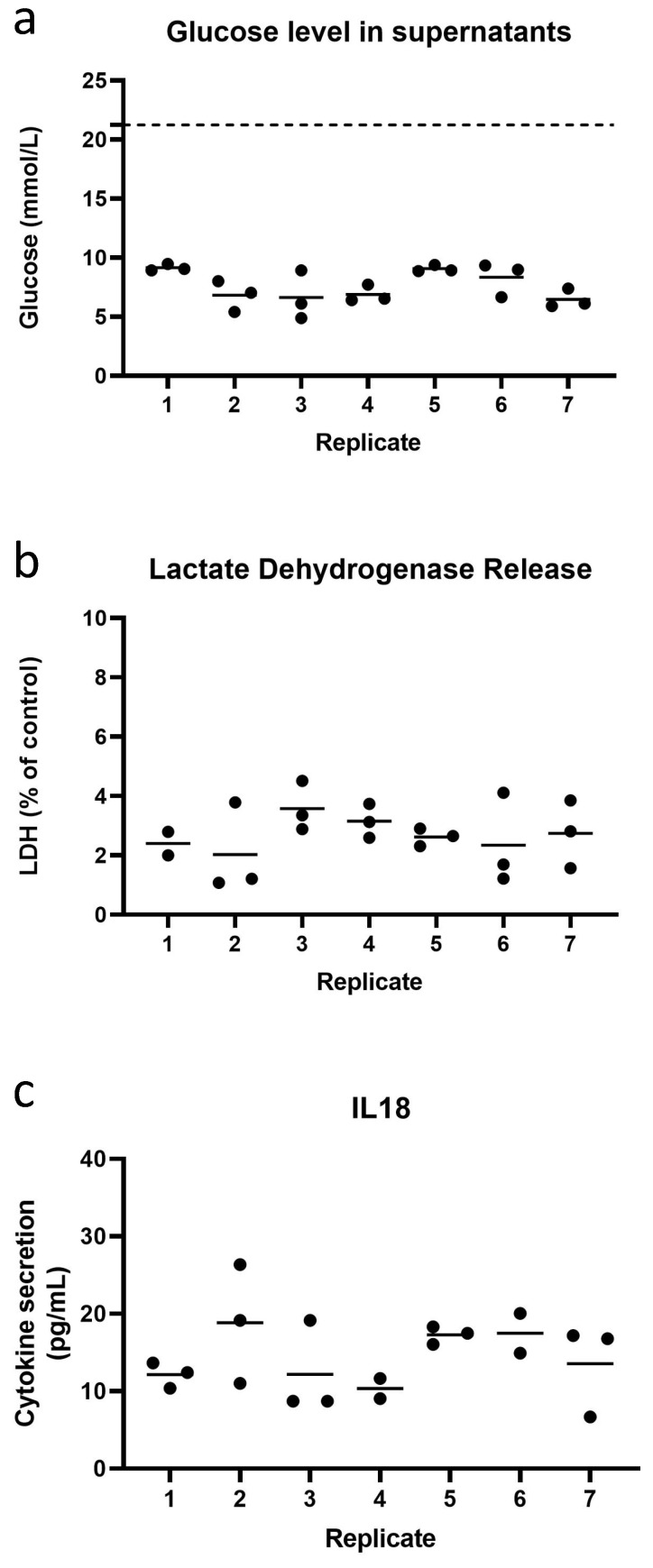
The assessment of intra-experiment reproducibility of RhS-NP cultured at day 10 under dynamic flow. Measurements of (**a**) glucose level (glucose concentration in medium is shown in a dashed line), (**b**) LDH and (**c**) cytokine IL-18 release in the supernatant of RhS-NP of single experimental repeat. Each black dot represents a single RhS-NP dynamic culture and cultures are already grouped in preparation for dose response chemical exposure. The horizontal line represents mean; ≤3 intra-experimental replicates. No significant difference was found with one-way ANOVA (Kruskal-Wallis test).

**Figure 5 pharmaceutics-14-01529-f005:**
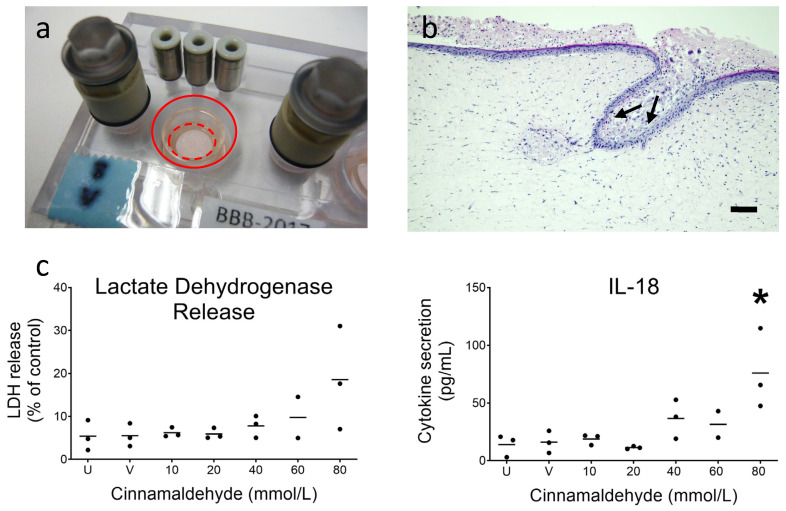
Topical exposure of cinnamaldehyde onto RhS-NP for 24 h starting at day 10 of dynamic flow culture. (**a**) Macroscopic view of topical application using filter paper (marked in red) on the stratum corneum of RhS-NP. (**b**) H&E staining of the RhS-NP model exposed to cinnamaldehyde (60 mmol/L) for 24 h. Histology shows the detrimental effects of cinnamaldehyde penetrating into the invaginated epidermis and NP disruption (cf. Figure 2b–d; unexposed RhS-NP). Extreme disruptions reaching the basal layer are shown by arrows, Scale bar = 100 µm. (**c**) LDH levels in the supernatant of unexposed (u), vehicle (v) and exposed RhS-NP show a dose dependent cytotoxicity after cinnamaldehyde exposure (24 h) and the increased release of sensitizer biomarker Interleukin-18 (IL-18). Each black dot represents a single RhS-NP. The horizontal line represents mean; ≤3 intra-experimental replicates. A Shapiro-Wilk test was used to assess normality, and no significant difference was found with one-way ANOVA (Kruskal-Wallis test) for LDH; a significant difference was found for IL-18 at 80 mmol/L cinnamaldehyde with one-way ANOVA (Kruskal-Wallis test) followed by Dunn’s multiple comparisons test, * *p* < 0.05.

**Figure 6 pharmaceutics-14-01529-f006:**
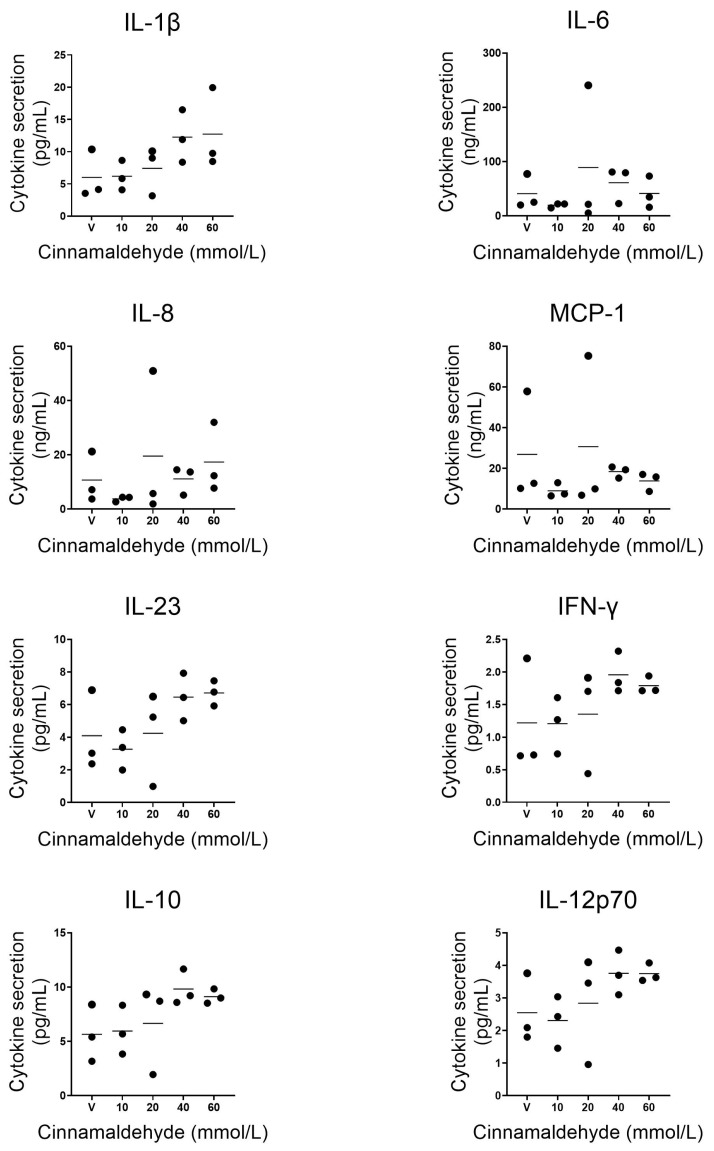
Cytokine release into culture supernatant after cinnamaldehyde exposure for 24 h starting at day 10 of dynamic flow culture. Pro-inflammatory cytokine IL-1β, inflammatory cytokines IL-6, IL-8 and MCP-1, inflammatory cytokines IL-23 and IFN-γ, and anti-inflammatory cytokines IL-10 and IL-12p70 are shown. Each dot represents a single RhS-NP. The horizontal line represents meanof three intra-experimental replicates. No significant differences were found with one-way ANOVA (Friedman test).

**Table 1 pharmaceutics-14-01529-t001:** Cytokine including chemokine baseline secretion from unexposed RhS-NP and function.

Cytokine	Unexposed RhS-NP, (Amount +/− SEM)	Function
**Pro-inflammatory**		
IL-18	14.0 +/− 5.5 pg/mL	Stimulates Th1 response; contact sensitizer specific biomarker
IL-33 #	<2.44 pg/mL	Member of IL-1 family; stimulates production of Th2 cytokines
IL-1β	7.6 +/− 2.1 pg/mL	Crucial for host-defence responses to infection and injury
TNF-α #	<0.42 pg/mL	Produced during acute inflammation, responsible for a diverse range of signaling events
**Inflammatory**		
IL-6	19.3 +/− 5.4 ng/mL	Stimulates Th17 response
IL-8/CXCL8	6.7 +/− 2.2 ng/mL	Potent chemoattractant for neutrophils, stimulates angiogenesis
IL17A #	<0.06 pg/mL	Mediates protective innate immunity to pathogens, contributes to inflammatory disease
IL-23	4.3 +/− 1.5 pg/mL	Maintains Th17 response; stimulates epidermal hyperplasia
IFN-α2 #	<0.32 pg/mL	Type I IFN; Inhibits cell proliferation; activates immune system; anti-viral; anti-tumor
IFN-γ	1.2 +/− 0.3 pg/mL	Type II IFN; stimulates innate and adaptive immunity
MCP-1/CCL2	15.4 +/− 4.8 ng/mL	Potant chemoatractant for monocytes and macrophages
**Anti-inflammatory**		
Il-10	6.8 +/− 2.0 pg/mL	Inhibits production of IFN-γ, IL-2, IL-3, TNF-α, GM-CSF by macrophages and Th1 cells
IL-12p70	2.9 +/− 0.8 pg/mL	Differentiation of naive T cells and Th1 cells; enhances cytotoxic activity of NK and CD8 cytotoxic lymphocytes; is required to induce IL-10, acts as an anti-inflamatory during secondary responses

Cytokine secretion between day 10 and day 11 (24 h) of unexposed reconstructed human skin with neopapillae (RhS-NP). Values are average +/− SEM of three replicates within a single repeat experiment used for cinnamaldehyde exposure. # cytokine secretion below the optimal detection limit of the assay before and after cinammaldehyde exposure. Detection limit of the assay is shown < indicates detection limit of assay.

## Data Availability

Data is contained within the article.

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
