# Peer review of "Proof-of-Concept Organ-on-Chip Study: Topical Cinnamaldehyde Exposure of Reconstructed Human Skin with Integrated Neopapillae Cultured under Dynamic Flow"

_pharmaceutics, 2022, doi:10.3390/pharmaceutics14081529_

Round 1
Reviewer 1 Report
Overall, it is a good paper with solid results.
Some comments:
1. In Figure. 2a, please add a plot showing the size distribution of the neopapillae spheroids.
2. In the lines 346-347, please comment on if it is possible to mitigate or even eliminate those outliners. If not, why?
Author Response
- In Figure. 2a, please add a plot showing the size distribution of the neopapillae spheroids.
Answer: In our previous study we fully characterized neopapillae diameter size showing a distribution curve of neopapillae diameter (see reference 23). As we are continuously preforming the same method for the neopapillae construction, we referred to this distribution curve of neopapillae diameter size as shown in our previous study and mentioned in the main text the mean diameter of mean diameter 97 µm; SEM ±16.6 µm to support the images in Figure 2a . see line 252-255.
- In the lines 346-347, please comment on if it is possible to mitigate or even eliminate those outliners. If not, why?
Answer: Since the focus of this study was to demonstrate intra- and inter- experiment variation these outliers were clearly represented in this study. However, upon further assay standardization, criteria could be reached to eliminate such outliers if necessary (e.g. if 2 out of 3 runs are consistent, the outlier may be eliminated from the final result interpretation if clearly indicated).
The text has been revised accordingly in lines 360 and 452-456
Reviewer 2 Report
This manuscript describes the integration of RhS with neopapillae in vitro models in a microfluidics bioreactor using the HUMIMIC Chip2 from TissUse. The authors have demonstrated static RhS-NP culture to dynamic flow with robust baseline data have been achieved within an experiment and between experiments.
A key achievement of this study is obtaining data from extensive intra-experimental and inter-experimental information on the viability and metabolic state of the cultures which I admire much.
I believe that the results demonstrated in this manuscript represent a step ahead toward realizing a robust skin model that can be integrated with downstream organoids for toxicity studies.
I support the publication of this manuscript after taking the following minor point into consideration:
§ The procedures for the Incorporation of RhS-NP into the HUMMIK Chip2 are not very clear. Authors may provide some more details including some explanatory drawings whenever possible. Also, figure 1 is somewhat blurred.
§ Why does the treatment of the RhS with the cinnamaldehyde need to be done off-chip? Can be performed in situ?
§ The re-constructed in vitro skin model is fragile and very sensitive to any mechanical stress. How did placing the filter paper impact the cell viability? was that done in air-liquid interface conditions?
§ Fig.2a: from where this image was taken? 96 wells?
§ Fig.5b: the disruption of the upper epidermal layers and the invaginating epidermis reaching towards the neopapillae should be highlighted on the image.
§ Figs 5 c and d: What do U and V on the x-axis indicate. This should be stated in the caption
§ I can’t see a comparison between the static and dynamic systems in terms of RhS characterization and robustness. The results obtained from the current study can be briefly compared with those from the previous static model. Any advantages of the dynamic miniaturized culture system other than the amenability to integration with a multi-organ system can be highlighted
§
Line 391 -398: incorporation of a vascularized network into the HUMIMIC microfluidic device is not part of the current study. This should be more clearly mentioned as it could be mistakenly considered otherwise.
Author Response
- The procedures for the Incorporation of RhS-NP into the HUMMIK Chip2 are not very clear. Authors may provide some more details including some explanatory drawings whenever possible. Also, figure 1 is somewhat blurred.
Answer: The text and figure has been revised accordingly.
See revised text lines 151-161 and Figure 1
- Why does the treatment of the RhS with the cinnamaldehyde need to be done off-chip? Can be performed in situ?
Answer: In the current study, the RhS-NP were indeed removed from the HUMMIC chip2 in order to ensure that the system could be thoroughly flushed with hydrocortisone free medium to avoid a possible hydrocortisone-induced immune suppression of cytokine release. We took this opportunity to topically apply the cinnamaldehyde before replacing RhS-NP into the chip. Whether this refreshment with hydrocortisone free medium is necessary has yet to be determined. If this is not, then in the future the RhS-NP could remain in the chip during chemical administration thus additional handlings in the future.
See revised discussion lines 463-470
- The re-constructed in vitro skin model is fragile and very sensitive to any mechanical stress. How did placing the filter paper impact the cell viability? was that done in air-liquid interface conditions?
Answer: The filter disks are applied to air exposed cultures to make sure that chemical penetration is via the stratum corneum and not via the hydrogel, thus mimicking the in vivo human patch test (line 323). Lactate dehydrogenase is only detected in culture supernatants when cell membranes become leaky due to cell toxicity. As observed in Figure 5c, LDH levels remain very low and comparable in both the unexposed (no filter paper) and vehicle (1% DMSO in water) impregnated filter paper disc indicating that addition of the filter paper disc and vehicle is not cytotoxic.
See revised text lines 186, and 330-334
- 2a: from where this image was taken? 96 wells?
Answer: The bulk neopapillae culture was grown in a 6 well plate.
See revised text (see line 123) and Figure 2 legend.
- 5b: the disruption of the upper epidermal layers and the invaginating epidermis reaching towards the neopapillae should be highlighted on the image.
Answer: Text has been revised to cf. Figure 2 b-d and Figure 5 line 329. Arrows have been added to the image highlighting disruption in invaginating epidermis and figure legend revised accordingly.
- Figs 5 c and d: What do U and V on the x-axis indicate. This should be stated in the caption
Answer: Legend has been revised accordingly
- I can’t see a comparison between the static and dynamic systems in terms of RhS characterization and robustness. The results obtained from the current study can be briefly compared with those from the previous static model. Any advantages of the dynamic miniaturized culture system other than the amenability to integration with a multi-organ system can be highlighted
Answer: We thank the reviewer for this comment. The future perspective has been expanded accordingly. See lines 482-493
- Line 391 -398: incorporation of a vascularized network into the HUMIMIC microfluidic device is not part of the current study. This should be more clearly mentioned as it could be mistakenly considered otherwise.
Answer: the text has been revised accordingly. See lines 407-408